# SINGLE-NODE ATTACK FOR FOOLING GRAPH NEURAL NETWORKS

## ABSTRACT

Graph neural networks (GNNs) have shown broad applicability in a variety of domains. Some of these domains, such as social networks and product recommendations, are fertile ground for malicious users and behavior. In this paper, we show that GNNs are vulnerable to the extremely limited scenario of a single-node adversarial example, where the node cannot be picked by the attacker. That is, an attacker can force the GNN to classify any target node to a chosen label by only slightly perturbing another *single arbitrary node* in the graph, even when *not being able to pick that specific attacker node*. When the adversary is allowed to *pick a specific attacker node*, the attack is even more effective. We show that this attack is effective across various GNN types (e.g., GraphSAGE, GCN, GAT, and GIN), across a variety of real-world datasets, and as a targeted and non-targeted attack. Our code is available anonymously at https://github.com/gnnattack/SINGLE.

## 1 INTRODUCTION

*Graph neural networks* (GNNs) (Scarselli et al., 2008; Micheli, 2009) have recently shown sharply increasing popularity due to their generality and computation-efficiency (Duvenaud et al., 2015; Li et al., 2016; Kipf & Welling, 2017; Hamilton et al., 2017; Veličković et al., 2018; Xu et al., 2019b). Graph-structured data underlie a plethora of domains such as citation networks (Sen et al., 2008), social networks (Leskovec & Mcauley, 2012; Ribeiro et al., 2017; 2018), knowledge graphs (Wang et al., 2018; Trivedi et al., 2017; Schlichtkrull et al., 2018), and product recommendations (Shchur et al., 2018). Therefore, GNNs are applicable for a variety of real-world structured data.

While most work in this field has focused on improving the accuracy of GNNs and applying them to a growing number of domains, only a few past works have explored the vulnerability of GNNs to adversarial examples. Consider the following scenario: a malicious user joins a social network such as Twitter or Facebook. The malicious user mocks the behavior of a benign user, establishes connections with other users, and submits benign posts. After some time, the user submits a new adversarially crafted post, which might seem irregular but overall benign. Since the GNN represents every user according to all the user's posts, this new post perturbs the representation of the user as seen by a GNN. As a result, another, specific benign user gets blocked from the network; alternatively, another malicious user submits a hateful post – but does not get blocked. This scenario is illustrated in Figure 1. In this paper, we show the feasibility of such a troublesome scenario: a single attacker node can perturb its own representation, such that another node will be misclassified as a label of the attacker's choice.

Most previous work on adversarial examples in GNNs required the perturbation to span *multiple* nodes, which in reality requires the cooperation of multiple attackers. For example, the pioneering work of Zügner et al. (2018) perturbed a *set* of attacker nodes; Bojchevski & Günnemann (2019a) perturb edges that are covered by a *set* of nodes. Further and in contrast with existing work, we show that perturbing a single *node* is more harmful than perturbing a single *edge*.

In this paper, we present a first a single-node adversarial attack on graph neural networks. If the adversary is allowed to choose the attacker node, for example, by hacking into an existing account, the efficiency of the attack significantly increases. We present two approaches for choosing the attacker: a white-box gradient-based approach, and a black-box, model-free approach that relies on graph topology. Finally, we perform a comprehensive experimental evaluation of our approach on multiple datasets and GNN architectures.

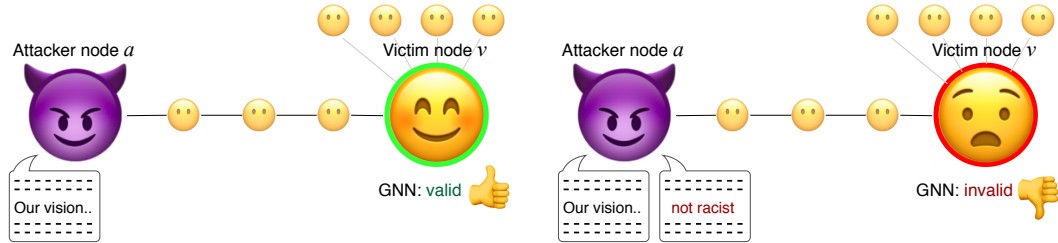

(a) Before attacking: the victim node ($v$) is classified as valid.

(b) After attacking: the victim node ($v$) is classified as invalid.

Figure 1: An partial adversarial example from the test set of the Twitter dataset. An adversarially-crafted post perturbs the representation of the attacker node. This perturbation causes a misclassification of the target victim node, although they are not even direct neighbors.

## 2  PRELIMINARIES

Let $\mathcal{G} = \{G_i\}_{i=1}^{N_G}$ be a set of graphs. Each graph $G = (\mathcal{V}, \mathcal{E}, \boldsymbol{X}) \in \mathcal{G}$ has a set of nodes $\mathcal{V}$ and a set of edges $\mathcal{E} \subseteq \mathcal{V} \times \mathcal{V}$, where $(u, v) \in \mathcal{E}$ denotes an edge from a node $u \in \mathcal{V}$ to a node $v \in \mathcal{V}$. $\boldsymbol{X} \in \mathbb{R}^{N \times D}$ is a matrix of $D$-dimensional node features. The $i$-th row of $\boldsymbol{X}$ is the feature vector of the node $v_i \in \mathcal{V}$ and is denoted as $\boldsymbol{x}_i = \boldsymbol{X}_{i,:} \in \mathbb{R}^D$.

**Graph neural networks** GNNs operate by iteratively propagating neural messages between neighboring nodes. Every GNN layer updates the representation of every node by aggregating its current representation with the current representations of its neighbors.

Formally, each node is associated with an initial representation $\boldsymbol{x}_v^{(0)} = \boldsymbol{h}_v^{(0)} \in \mathbb{R}^D$. This representation is considered as the given features of the node. Then, a GNN layer updates each node's representation given its neighbors, yielding $\boldsymbol{h}_v^{(1)} \in \mathbb{R}^{d_1}$ for every $v \in \mathcal{V}$. In general, the $\ell$-th layer of a GNN is a function that updates a node's representation by combining it with its neighbors:

$$\boldsymbol{h}_v^{(\ell)} = \text{COMBINE}\left(\boldsymbol{h}_v^{(\ell-1)}, \{\boldsymbol{h}_u^{(\ell-1)} \mid u \in \mathcal{N}_v\}; \theta_\ell\right), \tag{1}$$

where $\mathcal{N}_v$ is the set of direct neighbors of $v$: $\mathcal{N}_v = \{u \in \mathcal{V} \mid (u, v) \in \mathcal{E}\}$.

The COMBINE function is what mostly distinguishes GNN types. For example, graph convolutional networks (GCN) (Kipf & Welling, 2017) define a layer as:

$$\boldsymbol{h}_v^{(\ell)} = \text{ReLU}\left(\sum\nolimits_{u \in \mathcal{N}_v \cup \{v\}} \frac{1}{c_{u,v}} \boldsymbol{W}^{(\ell)} \boldsymbol{h}_u^{(\ell-1)}\right) \tag{2}$$

where $c_{u,v}$ is a normalization factor usually set to $\sqrt{|\mathcal{N}_v| \cdot |\mathcal{N}_u|}$. After $\ell$ such aggregation iterations, every node representation captures information from all nodes within its $\ell$-hop neighborhood. The total number of layers $L$ is usually determined empirically as a hyperparameter. In the node classification scenario, we use the final representation $\boldsymbol{h}_v^L$ to classify $v$.

For brevity, we focus our definitions on the semi-supervised transductive node classification goal, where the dataset contains a single graph $G$, and the split into training and test sets is across nodes in the same graph. Nonetheless, these definitions can be trivially generalized to the inductive setting, where the dataset contains multiple graphs, the split into training and test sets is between graphs, and the test nodes are unseen during training.

We associate each node $v \in \mathcal{V}$ with a class $y_v \in \mathcal{Y} = \{1, ..., Y\}$. The labels of the training nodes are given during training; the test nodes are seen during training – without their labels. The training subset is represented as $\mathcal{D} = \left(G, \{(v_i, y_i)\}_{i=0}^{N_\mathcal{D}}\right)$. Given the training set, the goal is to learn a model $f_\theta : (\mathcal{G}, \mathcal{V}) \to \mathcal{Y}$ that will classify the rest of the nodes correctly. During training, the model $f_\theta$ thus minimizes the loss over the given labels, using $J(\cdot, \cdot)$, which typically is the cross-entropy loss:

$$\theta^* = \operatorname{argmin}_\theta \mathcal{L}(f_\theta, \mathcal{D}) = \operatorname{argmin}_\theta \frac{1}{N_\mathcal{D}} \sum\nolimits_{i=0}^{N_\mathcal{D}} J\left(f_\theta\left(G, v_i\right), y_i\right) \tag{3}$$

## 3  SINGLE-NODE GNN ATTACK

In this section, we describe our *Single-node INdirect Gradient adversariaL Evasion* (*SINGLE*) attack. While our attack is simple, it is the first attack that focuses on perturbing nodes (in contrast to edges (Dai et al., 2018)), which works with an *arbitrary single attacker node* (in contrast to multiple nodes (Zügner et al., 2018)) that is not the node under attack (in contrast to "direct" attacks where the attacker perturbs the node under attack directly (Zügner et al., 2018; Li et al., 2020)).

### 3.1  PROBLEM DEFINITION

Given a graph $G$, a trained model $f_\theta$, a "victim" node $v$ from the test set along with its classification by the model $\hat{y}_v = f_\theta(G, v)$, we assume that an adversary controls another node $a$ in the graph. The goal of the adversary is to modify its own feature vector $x_a$ by adding a perturbation vector $\eta \in \mathbb{R}^D$ of its choice, such that the model's classification of $v$ will *change*.

We denote by $G_{x_a+\eta}$ the graph $G$ where the row of $X$ that corresponds to the node $a$ was added with the vector $\eta$. In a non-targeted attack, the goal of the attacker is to find a perturbation vector $\eta$ that will change the classification to *any* other class, i.e., $f_\theta(G_{x_a+\eta}, v) \neq f_\theta(G, v)$. In a *targeted* attack, the adversary chooses a specific label $y_{adv} \in \mathcal{Y}$ and the adversary's goal is to force $f_\theta(G_{x_a+\eta}, v) = y_{adv}$.

Generally, the classification of a node $v$ depends only on nodes whose *distance* to $v$ in the graph is lower than or equal $L$ – the number of GNN layers. Thus, a modification of the features of $a$ will affect the classification of $v$ only if the *distance* between $a$ and $v$ is lower than or equal $L$. Otherwise, $a$ will not be contained in the receptive field of $v$, and the attack will result in "under-reaching" (Alon & Yahav, 2020) – any perturbation of $a$ will not affect the prediction of $v$ (Barceló et al., 2020). Therefore, we require that $\text{distance}_G(a, v) \leq L$.

In this work, we focus on gradient-based attacks. These kinds of attacks assume that the attacker can access a similar model to the model under attack and compute gradients. As recently shown by Wallace et al. (2020), this is reasonable assumption: an attacker can query the original model; using these queries, imitate the model under attack by training an imitation model; find adversarial examples using the imitation model; and transfer these adversarial examples back to the original model. Under this assumption, these attacks are general and are applicable to any GNN and dataset.

### 3.2  CHALLENGES

**Unnoticeable Perturbations.** Our first challenge is to find an adversarial example that will allow an imperceptible perturbation of the input. This objective is attainable in continuous domains such as images (Szegedy et al., 2013; Goodfellow et al., 2014) and audio (Carlini & Wagner, 2018) if we constrain $l_\infty$-norm of the perturbation vector $\eta$. It is, however, unclear what imperceptibility means in graphs. In most GNN datasets, a node's features are a bag-of-words representation of the words that are associated with the node. For example, in Cora (McCallum et al., 2000; Sen et al., 2008), every node is annotated by a many-hot feature vector of words that appear in the paper; in PubMed (Namata et al., 2012), node vectors are TF-IDF word frequencies; in Twitter (Ribeiro et al., 2017), node features are averages of GloVe embeddings, which can be viewed as word frequency vectors multiplied by a (frozen) embedding matrix. We argue that an attack would be unnoticeable in an academic paper or in a set of Tweets if the frequency of some words is slightly modified. For example, a particular word may be repeated a few times throughout the text or remain unused.

To constrain the $\eta$ vector, we require that $\|\eta\|_\infty \leq \epsilon_\infty$ – the maximal absolute value of the elements in the perturbation vector – is bounded by $\epsilon_\infty \in \mathbb{R}^+$.

**Perturbing nodes instead of edges.** Previous work mostly focused on perturbing graph *edges*. Zügner et al. (2018) perturb both edges and node features, but conclude that "perturbations in the structure lead to a stronger change in the surrogate loss compared to feature attacks"; Wu et al. (2019b) also conclude that "perturbing edges is more effective than modifying the features". In this paper, we counter these conclusions and show that small node feature perturbations are stronger: (i) first, removing all the edges of a particular node is a special case of node feature perturbation. There exists a perturbation $\eta$ such that $W^1(x_a + \eta) = 0$, i.e., the modified feature vector $x_a + \eta$ is in the null

space of the first GNN layer.[1] Such a feature perturbation is equivalent to *removing all the edges* of the node $a$. (ii) Second, we argue that perturbing the graph structure is not realistic, because a single attacker controls only its own edges, and cannot control the *global* graph structure as in previous work (Dai et al., 2018; Bojchevski & Günnemann, 2019b; Zhang & Zitnik, 2020). (iii) Finally, when a successful attack is caused by removing edges, it is unclear whether the misclassification is caused by sensitivity to non-robust features in the data (Ilyas et al., 2019), or simply due to smaller amount of information. Similarly, when a successful attack is caused by inserting edges, it is unclear whether this is simply due to incorrect or unrealistic added information.

### 3.3 FINDING THE PERTURBATION VECTOR

To find the perturbation, we iteratively differentiate the desired loss of $v$ with respect to the perturbation vector $\boldsymbol{\eta}$, update $\boldsymbol{\eta}$ according to the gradient, and add it to the feature vector. In non-targeted attacks, we take the positive gradient of the loss of the undesired label to increase the loss; in targeted attacks, we take the negative gradient of the loss of the adversarial label $\boldsymbol{y}_{adv}$:

$$\boldsymbol{\eta}^{t+1} = \begin{cases} \boldsymbol{\eta}^t + \gamma \nabla_{\boldsymbol{\eta}} J\left(f_\theta\left(G_{\boldsymbol{x}_a+\boldsymbol{\eta}^t}, v\right), \hat{y}_v\right) & \text{non-targeted attack} \\ \boldsymbol{\eta}^t - \gamma \nabla_{\boldsymbol{\eta}} J\left(f_\theta\left(G_{\boldsymbol{x}_a+\boldsymbol{\eta}^t}, v\right), y_{adv}\right) & \text{targeted attack} \end{cases} \quad (4)$$

where $\gamma \in \mathbb{R}^+$ is a learning rate. We repeat this process for a predefined number of $K$ iterations, or until the model predicts the desired label.

**Enforcing the constraints.** We treat the node features as continuous throughout the attack iterations, whether they are discrete or continuous. Once the attack succeeds, we try to reset to zero as many perturbation vector elements as possible. We sort the perturbation vector elements in a decreasing order, according to their absolute value: $i_1, ..., i_D$. We start with the index of $\boldsymbol{\eta}$ whose absolute value is the largest, $\boldsymbol{\eta}_{i_1}$, and reset the rest of the $\{i_2, ..., i_D\}$ elements to zero. We then check whether perturbing only the $i_1$ index is sufficient. If the attack succeeds, we stop. If the attack fails (because of the large number of perturbation vector elements set to zero), we continue perturbing the rest of the elements of $\boldsymbol{\eta}$. In the worst case, we perturb all $D$ vector elements of $\boldsymbol{\eta}$. In most cases, we stop much earlier, practically perturbing only a small fraction of the vector elements. If the original node features are discrete, we discretized features after the optimization.

**Differentiate by frequencies, not by embeddings.** When taking the gradient with respect to the perturbation vector $\nabla_{\boldsymbol{\eta}}$, there is a subtle, but crucial, difference between the way that node representations are given in the dataset: (a) *indicative* datasets provide initial node representations $\boldsymbol{X} = [\boldsymbol{x}_1, \boldsymbol{x}_2, ...]$ that are word indicator vectors (many-hot) or frequencies such as (weighted) bag-of-words (Sen et al., 2008; Shchur et al., 2018); (b) in *encoded* datasets, initial node representations are given encoded, e.g., as an average of word2vec vectors (Hamilton et al., 2017; Hu et al., 2020). *Indicative* datasets can be converted to *encoded* by multiplying every vector by an embedding matrix; *encoded* datasets *cannot* be converted to *indicative*, without the authors releasing the textual data that was used to create the *encoded* dataset.

In *indicative* datasets, a perturbation of a node vector *can* be realized as a perturbation of the original text from which the *indicative* vector was derived. That is, adding or removing words in the text can result in the perturbed node vector. In contrast, a few-indices perturbation in *encoded* datasets might be an effective attack, but will *not* be realistic because there is no perturbation of the original text that will result in that perturbation of the vector. That is, when perturbing nodes, it is crucial to use *indicative* datasets, or convert *encoded* datasets to the *indicative* representation from which they were derived (as we do in Section 4) using their original text.

## 4 EVALUATION

We evaluate and analyze the effectiveness of our *SINGLE* attack. In Section 4.1, we show that *SINGLE* is more effective than alternatives such as single-edge attacks. In Section 4.2, we show that if we are allowed to *choose* the attacker node, *SINGLE* is significantly more effective.

**Setup.** Our implementation is based on PyTorch Geometric (Fey & Lenssen, 2019) and its provided datasets. We trained each GNN type with two layers ($L = 2$), using the Adam optimizer, early

---

[1]This equation demonstrates GCN, but similar equations hold for other GNN types like GAT and GIN.

|  | **Cora** | **CiteSeer** | **PubMed** | **Twitter** |
|---|---|---|---|---|
| Clean (no attack) | $80.5 \pm 0.8$ | $68.5 \pm 0.7$ | $78.5 \pm 0.6$ | $89.1 \pm 0.2$ |
| EdgeGrad | $65.1 \pm 1.3$ | $48.15 \pm 0.9$ | $59.7 \pm 0.7$ | $82.7 \pm 0.0$ |
| *SINGLE* | $\mathbf{60.1 \pm 0.1}$ | $\mathbf{34.0 \pm 3.6}$ | $\mathbf{45.5 \pm 0.5}$ | $\mathbf{72.1 \pm 7.2}$ |
| *SINGLE*-hops | $69.3 \pm 0.9$ | $45.1 \pm 5.2$ | $48.7 \pm 0.9$ | $74.5 \pm 6.7$ |

Table 1: Test accuracy (lower is better) under different types of attacks, when the attacker node is chosen *randomly*. Performed using GCN, $\epsilon_\infty = 1$ for the discrete datasets (Cora and CiteSeer), and $\epsilon_\infty = 0.1$ for the continuous datasets (PubMed and Twitter).

stopped according to the validation set, and applied a dropout of $0.5$ between layers. We used up to $K = 20$ attack iterations. All experiments in this section were performed with GCN, except for Section 4.5, where additional GNN types (GAT, GIN, and GraphSAGE) are shown. In Appendix A.2, we show consistent results across additional GNN types: GAT (Veličković et al., 2018), GIN (Xu et al., 2019b), GraphSAGE (Hamilton et al., 2017), SGC (Wu et al., 2019a), and RobustGCN (Zügner & Günnemann, 2019).

**Data.** We used Cora and CiteSeer (Sen et al., 2008) which are *discrete* datasets, i.e., the given node feature vectors are many-hot vectors. Thus, we set $\epsilon_\infty = 1$, the minimal possible perturbation. We also used PubMed (Sen et al., 2008) and the Twitter-Hateful-Users (Ribeiro et al., 2017) datasets, which are *continuous*, and node features represent frequencies of words. Continuous datasets allow a much more subtle perturbation, and we set $\epsilon_\infty = 0.1$. An analysis of these values is presented in Section 4.5.

The Twitter-Hateful-Users dataset is originally provided as an *encoded* dataset, where every node is an average of GloVe vectors (Pennington et al., 2014). We reconstructed this dataset using the original text from Ribeiro et al. (2017), to be able to compute gradients with respect to the weighted histogram of words, rather than the embeddings. We took the most frequent 10,000 words as node features, and used GloVe-Twitter embeddings to multiply by the node features. We thus converted this dataset to *indicative* rather than *encoded*. Statistics of all dataset are provided in the supplementary material.

**Baselines.** In *SINGLE* (Section 3.3) the attacker node is selected randomly for each victim node, and the attack perturbs this node's features according to $\epsilon_\infty$. SINGLE-*hops* is a modification of *SINGLE* where the attacker node is sampled *only among nodes that are not neighbors*, i.e., the attacker and the victim are not directly connected ($(a, v) \notin \mathcal{E}$). We compare to additional approaches from the literature: *EdgeGrad* follows most previous work (Xu et al., 2019a; Li et al., 2020; Zügner & Günnemann, 2020): *EdgeGrad* randomly samples an attacker node as in *SINGLE*, and either inserts or removes a single edge from or to the attacker node, according to the gradient.[2] If both use a randomly selected attacker node, *EdgeGrad* is strictly stronger than the *GradArgmax* attack of Dai et al. (2018), which only *removes* edges. We ran each approach 5 times with different random seeds for each dataset, and report the mean and standard deviation.

## 4.1 MAIN RESULTS

Table 1 shows our main results for non-targeted attacks across various datasets. As shown, *SINGLE* is more effective than *EdgeGrad* across all datasets. *SINGLE-hops*, which is more unnoticeable than attacking with a neighbor node, performs almost as good as *SINGLE* which attacks using a non-neighboring node, and better than *EdgeGrad*. On Twitter, *SINGLE* reduces the test accuracy significantly better than *EdgeGrad*: 72.1% compared to 82.7%. Results for *targeted* attacks are shown in Appendix A.3.

Surprisingly, Table A.5 shows that Robust GCN (Zügner & Günnemann, 2019) is as vulnerable to the SINGLE attack as a standard GCN, showing that there is still much room for novel ideas and improvements to the robustness of current GNNs.

As we explain in Section 3.3, *SINGLE* tries to find a perturbation vector in which the number of perturbed elements is minimal. We measured the number of vector elements that the attack had

---

[2]This can be implemented easily using *edge weights*: training the GNN with weights of 1 for existing edges, adding all possible edges with weights of 0, and taking the gradient with respect to the vector of weights.

|  | **Cora** | **CiteSeer** | **PubMed** | **Twitter** |
|---|---|---|---|---|
| GlobalEdgeGrad | **29.7** ± 2.4 | **11.9** ± 0.8 | 15.3 ± 0.4 | 82.7 ± 0.0 |
| *SINGLE+GradChoice* | 31.0 ± 1.9 | 19.0 ± 4.2 | 8.5 ± 1.2 | 7.0 ± 1.1 |
| *SINGLE+Topology* | 31.1 ± 1.2 | 18.1 ± 3.4 | **5.2** ± 0.1 | **6.6** ± 0.5 |

Table 2: Test accuracy when the adversary can *choose* the attacker node.

perturbed in practice. In PubMed, *SINGLE* used 76 vector elements on average, which are 15% of the elements in the feature vector. In Cora, *SINGLE* perturbed 717 elements on average, which are 50%. In CiteSeer, *SINGLE* used 1165 attributes on average, which are 31% of the features. In Twitter, *SINGLE* used 892 attributes on average, which are 9% of the features. In the experiments shown in Table 1, we used $\epsilon_\infty = 0.1$ in the continuous datasets (PubMed and Twitter). If we allow larger values of $\epsilon_\infty$, we can reduce the number of perturbed vector elements: using $\epsilon_\infty = 0.5$ requires perturbing only 3% of the attributes on average to achieve the same effectiveness; using $\epsilon_\infty = 1$ requires perturbing only 1.6% of the attributes on average to achieve the same effectiveness (in PubMed, where varying $\epsilon_\infty$ is meaningful).

## 4.2 ATTACKER CHOICE

If the attacker could *choose* its node, e.g., by hijacking an existing account in a social network, could they increase the effectiveness of the attack? We examine the effectiveness of two approaches for choosing the attacker node.

*Gradient Attacker Choice* (*GradChoice*) chooses the attacker node according to the largest gradient with respect to the node representations (for a non-targeted attack): $a^* = \operatorname{argmax}_{a_i \in \mathcal{V}} \|\nabla_{\boldsymbol{x}_i} J\left(f_\theta\left(G, v\right), \hat{y}_v\right)\|_\infty$. The chosen attacker node is never the victim node itself.

*Topological Attacker Choice* (*Topology*) chooses the attacker node according to topological properties of the graph. As an example, we choose the neighbor of the victim node $v$ with the smallest number of neighbors: $a^* = \operatorname{argmin}_{a \in \mathcal{N}_v} |\mathcal{N}_a|$. The advantage of this approach is that the attacker choice is *model-free*: if the attacker cannot compute gradients, they can at least choose the most harmful attacker node, and then perform the perturbation itself using other non-gradient approaches such as ones proposed by Waniek et al. (2018) and Chang et al. (2020).

To perform a fair comparison, we compare these approaches with *GlobalEdgeGrad*, which is similar to *EdgeGrad* that can insert or remove an edge, with the difference that the chosen edge can be chosen *from the entire graph*.

**Results.** Results for these attacker choice approaches are shown in Table 2. The main results are that choosing the attacker node significantly increases the effectiveness of the *SINGLE* attack: for example, in Twitter, from 72.1% (Table 1) to 6.6% test accuracy (Table 2).

In datasets where the given initial node features are continuous (PubMed and Twitter), *SINGLE+Topology* and *SINGLE+GradChoice* show similar results: on Twitter accuracy difference is less than 0.5%; on PubMed *SINGLE+Topology* outperforms *SINGLE+GradChoice* by ~3%, even though *SINGLE+Topology* is model-free. Both of those attacks are more efficient than *GlobalEdgeGrad*, showing the superiority of node perturbation over edge perturbation in the global view. In Appendix A.4, we show that allowing *GlobalEdgeGrad* to insert and remove *multiple* edges that belong to the same attacker node does *not* lead to a significant improvement.

Interestingly, *GradChoice* and *Topology* agree on the choice of attacker node for 50.3% of the nodes in Cora, 78.7% of the nodes in CiteSeer, 51.0% of the nodes in PubMed, and on 55.0% of the nodes in Twitter, showing that the node selection can sometimes be performed model-free.

In datasets where the initial node features are discrete (Cora and CiteSeer), i.e., many-hot vectors, *GlobalEdgeGrad* reduces the test accuracy *more* than *GradChoice* and *Topology*. We believe that the reason is the difficulty of two-step optimization in discrete datasets: for example, *GradChoice* needs to choose the node, and find the perturbation afterwards. Finding a perturbation for a discrete vector is more difficult than in continuous datasets, and the choice of the attacker node may not be optimal.

|                      | Cora           | CiteSeer       | PubMed         | Twitter       |
|----------------------|----------------|----------------|----------------|---------------|
| *SINGLE-two attackers* | **7.1** $\pm$ **0.5**  | **8.2** $\pm$ **0.2**  | 27.7 $\pm$ 0.2 | –             |
| *SINGLE-direct*      | 21.2 $\pm$ 2.5 | 13.8 $\pm$ 2.1 | **0.3** $\pm$ **0.1**  | 57.6 $\pm$ 8.7 |
| *SINGLE*             | 60.1 $\pm$ 0.1 | 18.1 $\pm$ 3.4 | 45.5 $\pm$ 0.5 | 72.1 $\pm$ 7.2 |

Table 3: Scenario ablation: test accuracy under different attacking scenarios.

|                      | Standard training | Adversarial training |
|----------------------|-------------------|----------------------|
| Clean (no attack)    | 78.5 $\pm$ 0.6    | 76.9 $\pm$ 0.6       |
| *SINGLE*             | 45.5 $\pm$ 0.5    | 58.5 $\pm$ 2.7       |
| *SINGLE*-hops        | 48.7 $\pm$ 0.9    | 62.1 $\pm$ 2.5       |
| *SINGLE+GradChoice*  | 8.5 $\pm$ 1.2     | 30.6 $\pm$ 6.8       |
| *SINGLE+Topology*    | 5.2 $\pm$ 0.1     | 21.1 $\pm$ 2.1       |
| *SINGLE-two attackers* | 27.7 $\pm$ 0.2  | 40.7 $\pm$ 3.4       |
| *SINGLE-direct*      | **0.3** $\pm$ **0.1**     | **4.6** $\pm$ **1.1**        |

Table 4: Test accuracy while attacking a model that was adversarially trained on PubMed, with different types of attacks.

### 4.3 SCENARIO ABLATION

The main scenario that we focus on in this paper is a *SINGLE* approach that always perturbs a *single* node, which is *not* the victim node ($a \neq v$). We now examine our *SINGLE* attack in other, easier but less realistic, scenarios: *SINGLE-two attackers* follows Zügner et al. (2018) and Zang et al. (2020), randomly samples *two* attacker nodes and perturbs their features using the same approach as *SINGLE*. *SINGLE-direct* perturbs the victim node directly (i.e., $a = v$), an approach that was found to be the most efficient by Zügner et al. (2018). Table 3 shows the test accuracy of these ablations. In Appendix A.5.3, we additionally experiment with more than two attacker nodes.

### 4.4 ADVERSARIAL TRAINING

In the previous sections, we studied the effectiveness of the *SINGLE* attack. In this section, we investigate to what extent can adversarial training (Madry et al., 2018) defend against *SINGLE*. For each *training* step and labeled training node, we perform $K_{\text{train}}$ adversarial steps to adversarially perturb another randomly sampled node, exactly as in *SINGLE*, but at training time. The model is then trained to minimize the original cross-entropy loss and the adversarial loss:

$$\mathcal{L}(f_\theta, \mathcal{D}) = \frac{1}{2N_\mathcal{D}} \sum_{i=0}^{N_\mathcal{D}} \left( J\left( f_\theta\left( G, v_i \right), y_i \right) + J\left( f_\theta\left( G_{x_{a_i} + \boldsymbol{\eta}_i}, v_i \right), y_i \right) \right). \tag{5}$$

The main difference from Equation (3) is the adversarial term $J\left( f_\theta\left( G_{x_{a_i} + \boldsymbol{\eta}_i}, v_i \right), y_i \right)$, where $a_i$ is the randomly sampled attacker for the node $v_i$. In every training step, we randomly sample a new attacker for each victim node and compute new $\boldsymbol{\eta}_i$ vectors. After the model is trained, we attack the model with $K_{\text{test}}$ *SINGLE* adversarial steps. This is similar to Feng et al. (2019) and Deng et al. (2019), except that they used adversarial training as a regularizer, to improve the accuracy of a model while not under attack. In contrast, we use adversarial training to defend a model against an attack at test time. We used $K_{\text{train}} = 5$, as we found it to be the maximal value for which the model's accuracy is not significantly hurt while not under attack ("clean"), and $K_{\text{test}} = 20$ as in the previous experiments. As shown in Table 4, adversarial training indeed improves the model's robustness against the different *SINGLE* attacks. However, the main result of this section is that *SINGLE*, *SINGLE+GradChoice* and *SINGLE+Topology* are still very effective attacks, as they succeed in attacking the adversarially trained model, reducing its test accuracy to 58.5%, 30.6% and 21.1%, respectively.

### 4.5 SENSITIVITY TO $\epsilon_\infty$

How does the intensity of the adversarial perturbation affect the performance of the attack? Intuitively, we say that the less we restrict the perturbation (i.e., larger values of $\epsilon_\infty$), the more powerful the attack. We examine whether this holds in practice.

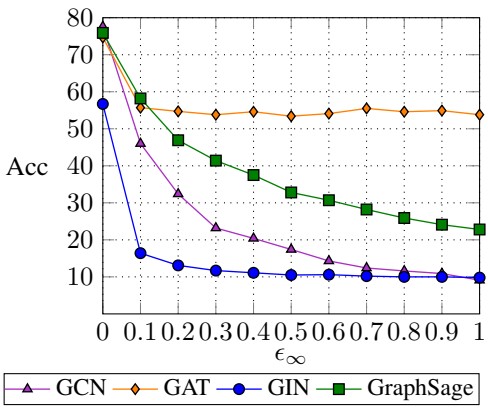

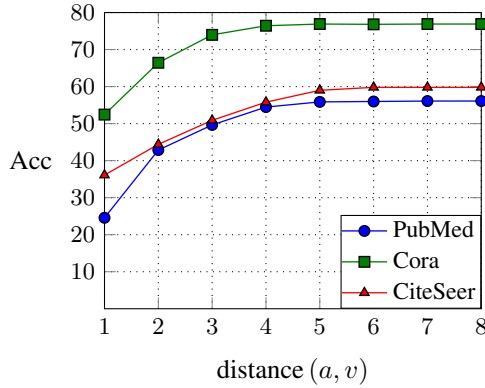

Figure 2: Effectiveness of the attack compared to the allowed $\epsilon_\infty$ (performed on PubMed, because its features are continuous).

Figure 3: Test accuracy compared to the distance between the attacker and the victim, when the GCN was trained with $L = 8$ on PubMed.

In our experiments in Sections 4.1 to 4.4, we used $\epsilon_\infty = 0.1$ for the continuous datasets (PubMed and Twitter). In this section, we vary the value of $\epsilon_\infty$ across different GNN types and observe the effectiveness of the attack. Figure 2 shows the results on PubMed. We used this dataset because it is larger than Cora and CiteSeer (Appendix A.1), and most importantly, its features are continuous, thus real-valued perturbations are feasible. As shown in Figure 2, the most significant difference is between performing the perturbation ($\epsilon_\infty = 0.1$) and not attacking at all ($\epsilon_\infty = 0$). As we increase the value of $\epsilon_\infty$, GCN and GraphSage (Hamilton et al., 2017) show a natural descent in test accuracy. Contrarily, GAT (Veličković et al., 2018) and GIN (Xu et al., 2019b) are more robust to increased absolute values of perturbations, while GAT is also the most robust compared to the other GNN types.

## 4.6 DISTANCE BETWEEN ATTACKER AND VICTIM

In Section 4.1, we found that *SINGLE* performs similarly to *SINGLE*-hops, although *SINGLE*-hops samples an attacker node $a$ whose distance from the victim node $v$ is at least 2. We further question whether the effectiveness of the attack depend on the distance in the graph between the attacker and the victim. We trained a new model for each dataset using $L = 8$ layers. Then, for each test victim node, we sampled attackers according to their distance to the test node.

As shown in Figure 3, the effectiveness of the attack increases as the distance between the attacker and the victim decreases. At distance of 5, the curve seems to saturate. A possible explanation for this is that apparently more than few layers (e.g., $L = 2$ in Kipf & Welling (2017)) are not needed in most datasets. Thus, the rest of the layers can theoretically learn *not* to pass much of their input starting from the redundant layers, excluding adversarial signals as well.

## 5 RELATED WORK

Works on adversarial attacks on GNN differ in several main aspects. In this section, we discuss the main criteria, to clarify the settings that we address.

**Single vs. multiple attackers** All previous works allowed perturbing multiple nodes, or edges that are covered by multiple nodes: Zügner et al. (2018) perturb features of a *set* of attacker nodes; Zang et al. (2020) assume "a few bad actors"; other works perturb edges that in realistic settings their perturbation would require controlling multiple nodes (Bojchevski & Günnemann, 2019a; Sun et al., 2020; Chen et al., 2018).

**Node vs. edge perturbations** Most adversarial attacks on GNNs perturb the input graph by modifying the graph *structure* (Zügner & Günnemann, 2019; Wang et al., 2020; Xu et al., 2019a). For example, Dai et al. (2018) iteratively remove edges, yet their attack manages to reduce the accuracy by about 10% at most when perturbing a single edge. Li et al. (2020) also allow the insertion of edges; Waniek

et al. (2018) and Chang et al. (2020) allow insertion and deletion of edges, using attacks that are based on correlations and eigenvalues, and not on gradients. Yefet et al. (2019) perturb one-hot node vectors, in the restricted domain of computer programs. Zügner et al. (2018) and Wu et al. (2019b) perturb both edges and nodes; but they concluded that perturbing edges is more effective than perturbing nodes. In this work, we counter these conclusions and show that perturbing *node* features is more effective than perturbing edges.

**Direct vs. influence attacks** Another difference between prior works lies in the difference between *direct attacks* and *influence attacks*. In direct attacks, the attacker *perturbs the target node itself*. For example, the attack of Zügner et al. (2018) is the most effective when *the attacker and the target are the same node*. In influence attacks, the perturbed nodes are at least one hop away from the victim node. In this paper, we show that the strong *direct* assumption is not required (*SINGLE-direct* in Section 4.2), and that our attack is effective *when the attacker and the target are not even direct neighbors*, i.e., they are at least *two* hops away (*SINGLE-hops* in Section 4.1).

**Poisoning vs. evasion attacks** In a related scenario, some work (Zügner & Günnemann, 2019; Bojchevski & Günnemann, 2019a; Li et al., 2020; Zhang & Zitnik, 2020) focuses on *poisoning* attacks that perturb examples *before* training. Contrarily, we focus on the standard *evasion* scenario of adversarial examples in neural networks (Szegedy et al., 2013; Goodfellow et al., 2014), where the attack operates at test time, *after* the model was trained, as Dai et al. (2018).

**Attacking vs. certifying** Zügner & Günnemann (2020) focus on *certifying* the robustness of GNNs against adversarial perturbations; and Bojchevski & Günnemann (2019b) certified PageRank-style models. In contrast, we study the effectiveness of the *adversarial attack* itself.

## 6 CONCLUSION

We demonstrate that GNNs are susceptible even to the extremely limited scenario of a single-node indirect adversarial example (*SINGLE*). The practical consequences of these findings are that a single attacker in a network can force a GNN to classify any *other* target node as the attacker's chosen label, by slightly perturbing some of the attacker's features. We further show that if the attacker can choose its attacker node – the effectiveness of the attack increases significantly. We study the effectiveness of these attacks across various GNN types and datasets.

We believe that this work will drive research in this field toward exploring novel defense approaches for GNNs. Such defenses can be crucial for real-world systems that are modeled using GNNs. Furthermore, we believe that the surprising results of this work motivate better theoretical understanding of the expressiveness and generalization of GNNs. To these ends, we make all our code and trained models publicly available.

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

# A SUPPLEMENTARY MATERIAL

## A.1 DATASET STATISTICS

Statistics of the datasets are shown in Table A.1.

Table A.1: Dataset statistics.

|  | #Training | #Val | #Test | #Unlabeled Nodes | #Classes | Avg. Node Degree |
|---|---|---|---|---|---|---|
| Cora | 140 | 500 | 1000 | 2708 | 7 | 3.9 |
| CiteSeer | 120 | 500 | 1000 | 3327 | 6 | 2.7 |
| PubMed | 60 | 500 | 1000 | 19717 | 3 | 4.5 |
| Twitter | 4474 | 248 | 249 | 95415 | 2 | 45.6 |

## A.2 ADDITIONAL GNN TYPES

Tables A.2 to A.4 present the test accuracy of different attacks applied on GAT (Veličković et al., 2018), GIN (Xu et al., 2019b), GraphSAGE (Hamilton et al., 2017), RobustGCN (Zügner & Günnemann, 2019), and SGC (Wu et al., 2019a), showing the effectiveness of *SINGLE* across different GNN types.

|  | Cora | CiteSeer | PubMed |
|---|---|---|---|
| EdgeGrad | $66.4 \pm 1.2$ | $49.4 \pm 1.4$ | $64.9 \pm 1.0$ |
| *SINGLE* | $\mathbf{40.0} \pm \mathbf{12.5}$ | $\mathbf{33.2} \pm \mathbf{6.7}$ | $35.7 \pm 13.3$ |
| *SINGLE*-hops | $42.0 \pm 11.5$ | $41.7 \pm 5.8$ | $\mathbf{35.5} \pm \mathbf{13.6}$ |
| GlobalGradEdge | $67.8 \pm 4.9$ | $48.3 \pm 5.1$ | $63.5 \pm 4.6$ |
| *SINGLE+GradChoice* | $43.1 \pm 4.9$ | $32.4 \pm 4.7$ | $36.4 \pm 8.0$ |
| *SINGLE+Topology* | $\mathbf{32.2} \pm \mathbf{6.4}$ | $\mathbf{25.5} \pm \mathbf{8.0}$ | $\mathbf{27.8} \pm \mathbf{5.7}$ |
| *SINGLE-two attackers* | $12.7 \pm 6.3$ | $\mathbf{11.0} \pm \mathbf{1.4}$ | $26.8 \pm 11.8$ |
| *SINGLE-direct* | $\mathbf{23.6} \pm \mathbf{1.5}$ | $14.8 \pm 4.3$ | $\mathbf{21.8} \pm \mathbf{2.5}$ |

Table A.2: Test accuracy of GAT under different non-targeted attacks

|  | Cora | CiteSeer | PubMed |
|---|---|---|---|
| EdgeGrad | $32.9 \pm 3.1$ | $18.5 \pm 3.0$ | $33.3 \pm 1.7$ |
| *SINGLE* | $\mathbf{27.1} \pm \mathbf{1.3}$ | $\mathbf{12.3} \pm \mathbf{2.9}$ | $\mathbf{12.9} \pm \mathbf{1.0}$ |
| *SINGLE*-hops | $32.6 \pm 0.7$ | $18.5 \pm 3.1$ | $14.0 \pm 0.6$ |
| GlobalGradEdge | $\mathbf{10.7} \pm \mathbf{2.8}$ | $\mathbf{4.8} \pm \mathbf{2.1}$ | $10.3 \pm 1.0$ |
| *SINGLE+GradChoice* | $15.9 \pm 2.0$ | $8.1 \pm 1.7$ | $10.0 \pm 1.6$ |
| *SINGLE+Topology* | $16.1 \pm 1.7$ | $7.6 \pm 1.6$ | $\mathbf{6.3} \pm \mathbf{1.7}$ |
| *SINGLE-two attackers* | $\mathbf{2.7} \pm \mathbf{0.7}$ | $5.4 \pm 2.0$ | $6.2 \pm 1.8$ |
| *SINGLE-direct* | $5.7 \pm 1.4$ | $\mathbf{4.7} \pm \mathbf{1.6}$ | $\mathbf{3.1} \pm \mathbf{3.1}$ |

Table A.3: Test accuracy of GIN under different non-targeted attacks

Surprisingly, Table A.5 shows that Robust GCN (Zügner & Günnemann, 2019) is as vulnerable to the SINGLE attack as a standard GCN, showing that there is still much room for novel ideas and improvements to the robustness of current GNNs.

## A.3 TARGETED ATTACKS

Tables A.6 to A.9 show the results of *targeted* attacks across datasets and approaches. Differently from other tables which show test accuracy, Tables A.6 to A.9 present the targeted attack's *success rate*, which is the fraction of test examples that the attack managed to force *a specific label prediction*

|  | Cora | CiteSeer | PubMed |
|---|---|---|---|
| EdgeGrad | $62.9 \pm 1.9$ | $45.9 \pm 3.4$ | $64.2 \pm 1.6$ |
| *SINGLE* | $\mathbf{62.7 \pm 2.4}$ | $\mathbf{32.3 \pm 4.3}$ | $\mathbf{57.1 \pm 0.8}$ |
| *SINGLE*-hops | $70.0 \pm 3.3$ | $45.5 \pm 4.3$ | $60.9 \pm 0.8$ |
| GlobalGradEdge | $48.9 \pm 2.7$ | $40.4 \pm 3.3$ | $64.7 \pm 1.1$ |
| *SINGLE+GradChoice* | $\mathbf{37.3 \pm 3.4}$ | $\mathbf{18.0 \pm 3.2}$ | $8.2 \pm 0.7$ |
| *SINGLE+Topology* | $37.4 \pm 3.6$ | $19.2 \pm 4.2$ | $\mathbf{6.6 \pm 0.3}$ |
| *SINGLE-two attackers* | $\mathbf{14.4 \pm 0.9}$ | $\mathbf{11.1 \pm 0.1}$ | $45.4 \pm 0.8$ |
| *SINGLE-direct* | $19.6 \pm 2.1$ | $13.5 \pm 3.9$ | $\mathbf{0.0 \pm 0.1}$ |

Table A.4: Test accuracy of GraphSAGE under different non-targeted attacks

|  | GCN | RobustGCN | SGC |
|---|---|---|---|
| Clean | $78.5 \pm 0.6$ | $73.9 \pm 1.6$ | $78.9 \pm 0.5$ |
| EdgeGrad | $59.7 \pm 0.7$ | – | $65.1 \pm 1.3$ |
| *SINGLE* | $\mathbf{45.5 \pm 0.5}$ | $34.3 \pm 1.4$ | $\mathbf{47.3 \pm 1.2}$ |
| *SINGLE+hops* | $48.7 \pm 0.9$ | $\mathbf{29.7 \pm 1.1}$ | $49.6 \pm 1.2$ |
| GlobalGradEdge | $15.3 \pm 0.4$ | – | $15.3 \pm 0.4$ |
| *SINGLE+GradChoice* | $8.5 \pm 1.2$ | $\mathbf{19.6 \pm 0.9}$ | $11.3 \pm 1.2$ |
| *SINGLE+Topology* | $\mathbf{5.2 \pm 0.1}$ | $72.5 \pm 1.9$ | $\mathbf{5.6 \pm 0.5}$ |
| *SINGLE+two attackers* | $27.7 \pm 0.2$ | $20.0 \pm 1.1$ | $30.0 \pm 1.8$ |
| *SINGLE+direct* | $\mathbf{0.3 \pm 0.1}$ | $\mathbf{15.8 \pm 1.1}$ | $\mathbf{0.5 \pm 0.2}$ |

Table A.5: Test accuracy of GCN, Robust GCN (Zügner & Günnemann, 2019), and SGC (Wu et al., 2019a) under different non-targeted attacks, on PubMed.

|  | Cora | CiteSeer | PubMed | Twitter |
|---|---|---|---|---|
| EdgeGrad | $8.0 \pm 0.7$ | $14.8 \pm 0.5$ | $20.1 \pm 0.6$ | $12.6 \pm 2.5$ |
| *SINGLE* | $\mathbf{36.6 \pm 2.4}$ | $\mathbf{60.7 \pm 2.2}$ | $\mathbf{38.2 \pm 0.6}$ | $\mathbf{14.6 \pm 4.7}$ |
| *SINGLE*-hops | $33.6 \pm 2.3$ | $50.0 \pm 2.5$ | $35.2 \pm 1.6$ | $12.6 \pm 3.7$ |
| GlobalGradEdge | $59.4 \pm 0.9$ | $\mathbf{78.7 \pm 0.9}$ | $80.1 \pm 0.6$ | $13.0 \pm 2.2$ |
| *SINGLE+GradChoice* | $\mathbf{65.8 \pm 1.5}$ | $67.2 \pm 2.2$ | $43.5 \pm 1.6$ | $42.2 \pm 11.6$ |
| *SINGLE+Topology* | $57.3 \pm 2.1$ | $66.3 \pm 3.0$ | $\mathbf{90.4 \pm 0.3}$ | $\mathbf{55.4 \pm 9.4}$ |

Table A.6: Success rate (higher is better) of different *targeted* attacks on a GCN network.

(in these results, higher is better). These results suggest that in targeted attack settings, node-based attacks (such as *SINGLE*) have an even bigger advantage over edge-based attacks (such as EdgeGrad).

|  | Cora | CiteSeer | PubMed |
|---|---|---|---|
| EdgeGrad | $6.1 \pm 0.4$ | $12.5 \pm 1.2$ | $17.9 \pm 1.5$ |
| *SINGLE* | $\mathbf{33.7 \pm 8.6}$ | $\mathbf{43.5 \pm 11.1}$ | $\mathbf{50.7 \pm 15.8}$ |
| *SINGLE*-indirect | $26.6 \pm 7.3$ | $29.88 \pm 8.6$ | $50.3 \pm 15.7$ |
| GlobalGradEdge | $6.0 \pm 1.4$ | $14.6 \pm 2.8$ | $22.3 \pm 3.6$ |
| *SINGLE+GradChoice* | $25.8 \pm 5.3$ | $38.6 \pm 8.5$ | $50.5 \pm 13.5$ |
| *SINGLE+Topology* | $\mathbf{41.3 \pm 5.3}$ | $\mathbf{52.5 \pm 11.3}$ | $\mathbf{63.0 \pm 10.2}$ |

Table A.7: Success rate (higher is better) of different *targeted* attacks on a GAT network.

|  | Cora | CiteSeer | PubMed |
|---|---|---|---|
| EdgeGrad | $16.8 \pm 1.2$ | $25.6 \pm 1.0$ | $37.9 \pm 2.6$ |
| *SINGLE* | $\mathbf{31.1} \pm \mathbf{1.7}$ | $\mathbf{49.0} \pm \mathbf{5.4}$ | $\mathbf{58.8} \pm \mathbf{7.9}$ |
| *SINGLE*-hops | $24.5 \pm 1.2$ | $37.4 \pm 4.1$ | $57.8 \pm 5.7$ |
| GlobalGradEdge | $44.7 \pm 4.7$ | $55.0 \pm 7.0$ | $64.9 \pm 11.8$ |
| *SINGLE+GradChoice* | $44.3 \pm 5.0$ | $\mathbf{59.0} \pm \mathbf{4.5}$ | $63.5 \pm 9.9$ |
| *SINGLE+Topology* | $45.1 \pm 2.3$ | $58.7 \pm 5.1$ | $\mathbf{73.2} \pm \mathbf{13.3}$ |

Table A.8: Success rate (higher is better) of different *targeted* attacks on a GIN network.

|  | Cora | CiteSeer | PubMed |
|---|---|---|---|
| EdgeGrad | $7.6 \pm 0.3$ | $16.3 \pm 1.7$ | $19.1 \pm 1.4$ |
| *SINGLE* | $\mathbf{24.3} \pm \mathbf{1.9}$ | $\mathbf{50.0} \pm \mathbf{2.5}$ | $\mathbf{27.9} \pm \mathbf{1.0}$ |
| *SINGLE*-hops | $15.4 \pm 3.8$ | $34.2 \pm 2.6$ | $24.1 \pm 1.0$ |
| GlobalGradEdge | $9.3 \pm 0.9$ | $14.7 \pm 1.1$ | $19.6 \pm 0.8$ |
| *SINGLE+GradChoice* | $49.1 \pm 3.0$ | $63.7 \pm 4.0$ | $36.3 \pm 2.1$ |
| *SINGLE+Topology* | $\mathbf{54.1} \pm \mathbf{1.3}$ | $\mathbf{69.3} \pm \mathbf{3.2}$ | $\mathbf{89.8} \pm \mathbf{0.3}$ |

Table A.9: Success rate (higher is better) of different *targeted* attacks on a GraphSAGE network.

## A.4 MULTIEDGE ATTACKS

We strengthened the *EdgeGrad* attack by allowing it to add and remove multiple edges that are connected to the attacker node – *MultiEdgeGrad*. Accordingly, *MultiGlobalEdgeGrad* is equivalent to *GlobalEdgeGrad*, except that *MultiGlobalEdgeGrad* can choose the attacker node.

|  | PubMed |
|---|---|
| Clean | $78.5 \pm 0.6$ |
| EdgeGrad | $65.1 \pm 1.3$ |
| MultiEdgeGrad | $64.5 \pm 0.2$ |
| *SINGLE* | $\mathbf{45.5} \pm \mathbf{0.5}$ |
| *SINGLE-hops* | $48.7 \pm 0.9$ |
| *SINGLE+Topology* | $\mathbf{5.2} \pm \mathbf{0.1}$ |
| *SINGLE+GradChoice* | $8.5 \pm 1.2$ |
| GlobalGradEdge | $15.3 \pm 0.4$ |
| MultiGlobalGradEdge | $15.3 \pm 0.5$ |

Table A.10: Test accuracy of GCN using MultiEdge attacks

As shown in Table A.10, allowing the attacker node to add and remove multiple edges (*MultiEdge-Grad* and *MultiGlobalEdgeGrad*) results in a very minor improvement compared to *EdgeGrad* and *GlobalEdgeGrad*, while *SINGLE*, *SINGLE+Topology* and *SINGLE+GradChoice* are much more effective.

## A.5 ADDITIONAL BASELINES

### A.5.1 ZERO-FEATURES APPROACH

We experimented with a baseline where we set $\eta = -x_a$ as the feature perturbation. The objective of experimenting with such an attack is to illustrate that *SINGLE* can find better perturbations than simply canceling the node feature vector, making the new vector a vector of zeros (and thus effectively removes the edges of the attacker node in GCN).

As shown, *Zero features* is barely effective (compared to "Clean"), and *SINGLE* can find much better perturbations.

| | PubMed |
|---|---|
| Clean | $78.5 \pm 0.6$ |
| *SINGLE* | $\mathbf{45.5 \pm 0.5}$ |
| Zero features | $76.6 \pm 0.3$ |

Table A.11: Test accuracy of our zero features attack on a GCN network.

### A.5.2 INJECTION ATTACKS

We also study an additional type of a realistic attack that is based on node injection. In this approach, we insert a new node to the graph with a single edge attached to our victim node. The attack is performed by perturbing the injected node's attributes. Since there is no initial node feature vector to measure the $\epsilon_\infty$ distance to, the injected node is allowed to find any realistic representation (e.g., without choosing negative frequencies). This attack is very powerful, reducing the test accuracy down to 0.02% on PubMed.

### A.5.3 LARGER NUMBER OF ATTACKERS

| Number of attackers | PubMed |
|---|---|
| 1 | 45.5 |
| 2 | 27.7 |
| 3 | 19.0 |
| 4 | 15.3 |
| 5 | 12.9 |

Table A.12: Test accuracy for different number of attackers on PubMed.

We performed additional experiments with up to five randomly sampled attacker nodes simultaneously (Table A.12). As expected, allowing a larger number of attackers reduces the test accuracy. However, the main observation in this paper is that even a single attacker node is surprisingly effective.

### A.6 LIMITING THE ALLOWED $\epsilon_0$

In Section 4.5, we analyzed the effect of the value of $\epsilon_\infty$, that is, the maximal allowed perturbation in each vector attribute, on the performance of the attack. However, in datasets such as Cora and CiteSeer, the input features are *binary* (i.e., the input node vector is many-hot), so the possible perturbation to each vector element is only "flipping" its value from zero to one, or vice-versa. Thus, in these datasets, it is interesting to analyze the value of $\epsilon_0$, the maximal *number* of allowed perturbed vector elements, on the performance of the attack. In this case, measuring the $l_0$ norm is equivalent to measuring the $l_1$ norm: $\|\boldsymbol{\eta}\|_0 = \|\boldsymbol{\eta}\|_1$, and is proportional to the $l_2$ norm.

We performed experiments where we measured the test accuracy of the model while limiting the number of allowed perturbed vector elements. The results are shown in Figure A.1.

As shown, when $\epsilon_0 = 0$, no attack is allowed, and the test accuracy is equal to the "Clean" value of Table 1. When $\epsilon_0 = 100\%$, the results are equal to the *SINGLE* values of Table 1 – resulting in flipping 50% of the features on average in Cora, and 31% of the features on average in CiteSeer.

It is important to note that in practice, the average number of perturbed features is *much lower* than the maximal number of allowed features. For example, in CiteSeer, allowing 100% of the features results in *actually using only 31%* on average.

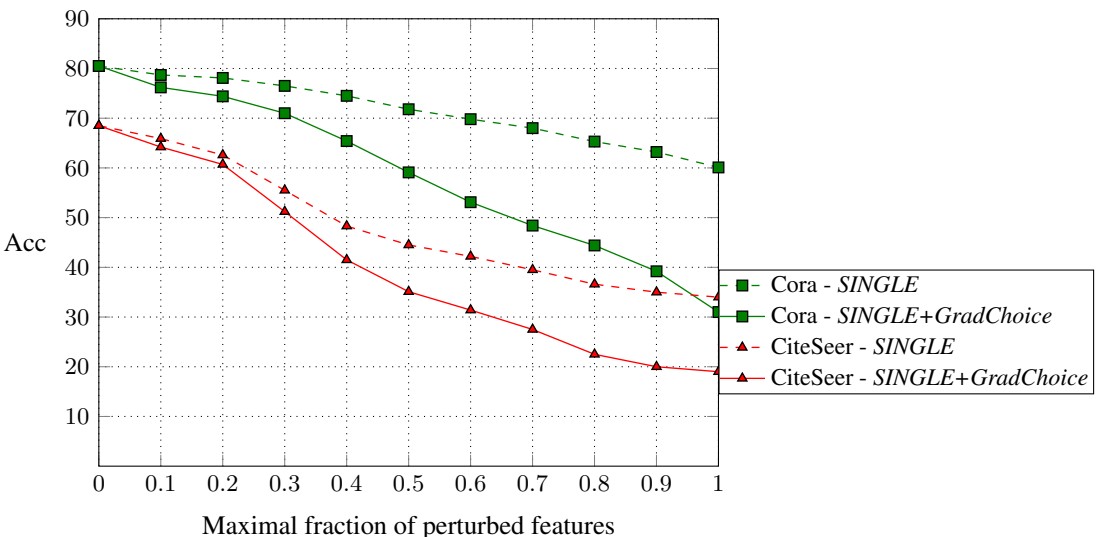

Figure A.1: Test accuracy compared to the maximal allowed $\epsilon_0$, the number of perturbed features (divided by the total number of features in the dataset). In practice, the average number of perturbed features is *much lower* than the maximal number of *allowed* features.

