# OpenReview forum: "Single-Node Attack for Fooling Graph Neural Networks"
_ICLR.cc/2021/Conference — Reject_

### Official Review · AnonReviewer1 · 2020-10-27
**ICLR 2021 Conference Paper421 AnonReviewer1**

**Rating:** 6
**Confidence:** 2

**Review:**

# summary #
This paper studies a problem of attacking graph neural networks.
Especially, the paper focuses on the single node attack while most of the attack studies focusing on edges.
The paper claims that attacks on multiple edges (and nodes) are difficult to conduct in practice.
Considering the single node attack is important since injecting a new single node, or a manipulation on one vulnerable node is less difficult.
The studied problem is an evasion attack on a node feature vector of a single node.
Untargeted and targetted attacks are both formulated with a gradient-based update of a perturbation vector. The proposed attack method perturbates BoW count vectors, not the embedded continuous vectors.
Experimental results show that the proposed single-node attack is more effective than a single-edge attack. Also, the single-node attack is not much weaker than the two-nodes attack experimentally.

# comment #
I am not an expert on the attacks of GNN. However, this manuscript is easy to read for non-expert readers like me.

Single-node interventions for GNNs sounds realistic than attacks on multiple edges.
I feel a node injection (add a new malicious node to a graph) is also a possible scenario but the paper does not consider this case. Any comments?

I find no obvious concerns about the methodology. However, I have a few questions concerning the experiments.

Table 1:
I'm not sure comparisons between the proposed SINGLE method and the baseline EdgeGrad method are fair comparisons, in terms of the amount of perturbations on graphs.
The proposed SINGLE method manipulates 15%, 50%, and 31% of the feature attributes for PubMed, Cora, and CiteSeer, respectively.
However, the counterpart EdgeGrad always manipulates a single edge. Is a single edge manipulation a (roughly) same amount of perturbations of MULTIPLE attribute manipulations of the SINGLE method?
An average degree (number of edges a node has) will be a good measure to check this issue. Is it possible to add an average degree to Table 4?

Table 3:
I'm curious about the differences between SINGLE-two attackers and the SINGLE.
The two-nodes attacks perform inferior 3 out of 4 datasets than the single-node attacks.
This conflicts with a naive guess: multi-node attacks is stronger than single-node attacks.
So I think many readers will expect some intuitive explanations. Gradients from two nodes harm each other in optimization? What will happen if we choose three, or more attackers?


# Evaluation points #
(+) Single-node evasion attack on GNN is first in the literature

(+) Readability

(-) I'm not fully sure about the fairness of parts of the experiment Table 1.

(-) No explanation in why two-nodes attacks are not superior to the one-node attacks.

---

> ### Author Response · Authors · 2020-11-18
> **Thank you for your review**
>
> Thank you for taking the time to review our paper!
> We were happy to read that the paper was easy to read and welcoming.
>
> Please see our detailed response below.
>
> > I feel a node injection (add a new malicious node to a graph) is also a possible scenario but the paper does not consider this case
>
> We implemented and experimented with *injecting* a single attacker that is a neighbor of the victim node. This allows to reduce the test accuracy **down to 2%** on PubMed.
>
> We will include this result in our next revision.
>
> > the counterpart EdgeGrad always manipulates a single edge.
>
> We strengthened the EdgeGrad attack in a new approach called MultiEdgeGrad. MultiEdgeGrad can add and remove *multiple* edges that are connected to the attacker node $a$. Accordingly, a new approach called MultiGlobalEdgeGrad is equivalent to GlobalEdgeGrad, except that MultiGlobalEdgeGrad can *choose* the attacker node, and then multiple edges to add and remove. Here are the results on PubMed, using GCN:
>
> |                   | When the attacker node is chosen *randomly*      |
> |-------------------|----------|
> | EdgeGrad          | 65.1     |
> | MultiEdgeGrad     | 64.5     |
> | SINGLE            | 45.5     |
> | SINGLE-hops       | 48.7     |
>
> As shown, allowing the attacker node to add and remove multiple edges (MultiEdgeGrad) results in a very minor improvement compared to EdgeGrad, while SINGLE is much more effective.
>
> |               | When the adversary can *choose* the attacker node |
> |-------------------|----------|
> | SINGLE+Topology   | 5.2      |
> | SINGLE+GradChoice | 8.5      |
> | GlobalEdgeGrad    | 15.3     |
> | MultiGlobalEdgeGrad | 15.3     |
>
> As shown, allowing the attacker node to choose the attacker node, and then add and remove multiple edges (MultiGlobalEdgeGrad) results in a very minor improvement compared to GlobalEdgeGrad, while SINGLE+Topology and SINGLE+GradChoice are much more effective.
>
> We will include these results in our next revision.
>
>
>
> > Is it possible to add an average degree to Table 4?
>
> |            | Cora    | CiteSeer    | PubMed    | Twitter
> |------------|------|------|------|------|
> | Avg. Degree | 3.9 | 2.7 | 4.5 |  45.6 |
>
> We will add these numbers in our next revision.
>
> >  The two-nodes attacks perform inferior 3 out of 4 datasets than the single-node attack... What will happen if we choose three, or more attackers?
>
> This was a mistake which we will fix in our next revision.
> Multi-node attacks are now included and results are attached below.
>
> |            | 1    | 2    | 3    | 4    | 5    |
> |------------|------|------|------|------|------|
> | No. of attackers | 45.5 | 29.2 | 19.0 | 15.3 | 12.9 |
>
> As expected, allowing a larger number of attackers reduces the test accuracy. However, the main observation in this paper is that even a *single* attacker node is surprisingly effective.
> We will include these results in our next revision.

---

> > ### Comment · AnonReviewer1 · 2020-11-19
> > **Thank you for your clarifications**
> >
> > Thank you for your author feedbacks.
> > The updated results of the multi-node attacks seems reasonable.
> >
> > I will read all reviews and feedbacks to re-consider my evaluations.

---

### Official Review · AnonReviewer4 · 2020-10-28
**Official Blind Review #4**

**Rating:** 6
**Confidence:** 5

**Review:**

This paper studies how to attack a specific attacker node by perturbing only one single node's feature. The authors proposed a gradient-based attack strategy, namely SINGLE, to learn the perturbation vector with the consideration of unnoticeability. In general, I really like the idea of this work. Compared with other attacking strategies, the attacking scenario in this paper is more realistic, e.g., the access to a single node (hacking the account), etc. And the gradient-based attack is intuitive and straightforward. Overall, this paper is well-written and easy to follow. I have the following concerns that would like the authors to clarify.

- In the paper, the authors claim that it can attacking by perturbing single arbitrary node. I think 'arbitrary' is a little bit over-claimed, since the distance between attacker node $a$ and the victim node $v$ should satisfy $d(a, v) \leq L$ (# layers).
- I am a little bit confused about how to process the encoded dataset. In section 4, the authors mention that the most frequent 10000 words are used as node features, then the glove embeddings are used to multiply by the embedding. Does it mean that the update rule in each layer becomes $\sigma(A X Z W)$, where $A$ = adjacency matrix (shape $n \times n$), $X$ = many-hot/TD-IDF encoding of 10000 words (shape $n \times 10000$), $Z$ = $d$-dimensional word embedding (shape $10000 \times d$) and $W$ = weight matrix (shape $d \times \textit{nhid}$)? If I understand it correctly, the derivative of loss wrt $X$ (i.e., many-hot/TD-IDF encoding) is calculated. If so, I think it could be better to make it clear in your manuscript.
- If the parameter $\epsilon_{\inf}$ is set to a very small value (like 0.1 or 1), it is possible that a lot of words in the original text would be perturbed. In this case, is it still reasonable to say the perturbation is unnoticeable? For example, in section 4.1, there are 50% and 31% of the features (I assume it means words) being perturbed. If 50% of words are perturbed in some texts, I think we can notice such changes as a human. A related but different question: what are the ratio of perturbed words if we set $\epsilon_{\inf}$ a little bit larger?
- A general question about your experiments: how does the test accuracy computed? If we only wish to attack one node, is it more reasonable to test the success rate (like waht the authors did in appendix). Or does it mean that the overall test accuracy can drop that much if we only perturb one node in the graph? This is a little bit unclear in the manuscript (or please point it out if there is something I missed).
- About the analysis on distance between attacker and victim nodes, it seems the test accuracy of cora and pubmed does not change once the # layers reaches to 5. How many different rounds of experiments do the authors conduct in this part? If an attacker node is randomly chosen, you cannot always find a victim node v that has a large $d(a, v)$. If the authors want to study the effect of distance, maybe the attacker node needs to be carefully chosen.
- In section 4.4, does the attacker still use GCN as base model (or imitation model) to attack GAT, GIN and GraphSage? If so, does the number of layers in imitation model remain the same as the attacked true model? What would happen if the number of layers in imitation model is different from the attacked model? I think all those models have some sort of nonlinearity baked in the model. Just for my curiosity, I wonder what would happen if we use a simpler model like SGC (without any nonlinearity) as the imitation model.
- Just some minor comments, I sometimes got confused about 'attacker node' and 'victim node'. It would be better if the authors can clearly define it somewhere in the paper or mark them (which one is which) on figure 1.

---

> ### Author Response · Authors · 2020-11-18
> **Thank you for your review**
>
> Thank you for taking the time to review our paper!
> We were happy to read that you "*really liked the idea of this work*" and that you agree with our observation that "*the attacking scenario in this paper is more realistic*".
> Please see our detailed response below.
>
> > the distance between attacker node a and the victim node v should satisfy  d(a,v)≤L
>
> This is a general limitation of GNNs - the classification of a node depends only on its neighborhood of radius L (the number of layers).
> Thus, attackers that their distance to the victim nodes is greater than the number of layers, are not considered by the victim node.
>
> >Does it mean that the update rule in each layer becomes $\sigma\left(AXZW\right)$? If so, I think it could be better to make it clear in your manuscript.
>
> Yes, the update rule is as you describe, and we differentiate with respect to the frequencies matrix $X$.
> We will clarify this in the revised version.
>
> > If 50% of words are perturbed in some texts, I think we can notice such changes as a human.
> >what are the ratio of perturbed words if we set $\epsilon_{\infty}$ a little bit larger?
>
> In *continuous datasets*, the percentage of perturbed words is much smaller: 15% in PubMed, and 9% in Twitter, while we limit the $\epsilon_{\infty}$ parameter to the small value of 0.1.
>
> If we allow using $\epsilon_{\infty} = 0.5$, this only requires perturbing 3% of the attributes on average to achieve the same effectiveness.
> If we allow using $\epsilon_{\infty} = 1$, this only requires perturbing 1.6% of the attributes on average to achieve the same effectiveness (in PubMed).
>
>
> >how does the test accuracy computed? If we only wish to attack one node, is it more reasonable to test the success rate (like what the authors did in appendix). Or does it mean that the overall test accuracy can drop that much if we only perturb one node in the graph?
>
> Test accuracy is computed exactly as in the original (unattacked model): the fraction of examples that are classified correctly, out of the test set. Every example in the test set is attacked separately, one node at a time.
>
> This means that the overall test accuracy can drop from the "Clean" value to the new test accuracy, if we perturb one node at a time, using a single attacker node at a time.
>
> In **targeted attacks** (in the Appendix) we reported the success rate of the attack, because there are cases when the correct label is $y_{true}$, the targeted attack aims to change the prediction to $y_{adv}$, but the model-under-attack predicts $w$. Such a case reduces the test accuracy (because $w \neq y_{true}$), but the targeted attack had failed (because $y_{adv} \neq w$). In such a case, we do *not* wish to measure this as a success, although the test accuracy was reduced.
>
>
> > it seems the test accuracy of cora and pubmed does not change once the # layers reaches to 5. How many different rounds of experiments do the authors conduct in this part? If an attacker node is randomly chosen, you cannot always find a victim node v that has a large d(a,v). If the authors want to study the effect of distance, maybe the attacker node needs to be carefully chosen.
>
> The test set contains 1000 nodes. In Figure 3, we thus attacked **each** of the test nodes with an attacker of every distance. Thus, each point in the figure is an average of 1000 different rounds.
>
>
> > In section 4.4, does the attacker still use GCN as base model (or imitation model) to attack GAT, GIN and GraphSage?
>
> In section 4.4 (Figure 2) - we trained 4 different models (GCN, GAT, GIN, GraphSAGE), and attacked each model directly (without an imitation model).
>
> > I wonder what would happen if we use a simpler model like SGC
>
> Here are the results:
>
> |                   | SGC - When the attacker node is chosen *randomly*      |
> |-------------------|----------|
> | Clean             | 78.9 |
> | EdgeGrad          | 65.1     |
> | SINGLE            | 47.3     |
> | SINGLE-hops       | 49.6     |
>
> As shown, SINGLE is more effective than EdgeGrad.
>
> |               | SGC - When the adversary can *choose* the attacker node |
> |-------------------|----------|
> | SINGLE+Topology   | 5.6      |
> | SINGLE+GradChoice | 11.3     |
> | GlobalEdgeGrad    | 15.3     |
>
> As shown, SINGLE+Topology and SINGLE+GradChoice are more effective than GlobalEdgeGrad    .
>
>
> >I sometimes got confused about 'attacker node' and 'victim node'. It would be better if the authors can clearly define it somewhere in the paper or mark them (which one is which) on figure 1.
>
> Thank you for pointing this out. We will clarify this in the next revision.

---

### Official Review · AnonReviewer3 · 2020-10-28
**Interesting insights, sufficient empirical results, but risks over-claiming**

**Rating:** 6
**Confidence:** 3

**Review:**

This submission shows that a single-node attack for GNNs can be surprisingly effective. The discovery is mostly in the form of experimental results, rather than in the form of a new method.

Strengths:

S1: The discovery that a single-node attack can be surprisingly effective is indeed interesting. It can potentially complement our understanding of GNNs and bring novel ideas in the future.

S2: The empirical results are mostly sufficient to support the main insight that a single-node attack can be effective in many cases. Still, it would be even better if Table 2 can include results for models other than the vanilla GNNs (e.g., GAT) (note that there aren’t GAT’s results on Cora, CiteSeer, Twitter, even though GAT’s result on PubMed is shown in Figure 2).

S3: Good literature review. Friendly to even the readers who don’t follow the field of adversarial attacks closely.

Weaknesses:

W1: The abstract is a bit exaggerating the main discovery. The empirical results only suggest that single-node attacks are effective under the following conditions: (1) the model is the vanilla GNN (while GAT is much harder to attack), (2) the attack node is close enough to the victim node (<= 3 hops), and (3) the node features are of the continuous kind (the results on datasets of discretized features are much weaker).

W2: The proposed method seems unrealistic for datasets of discretized (one-hot, multi-hot, integer-valued, etc.) node features. (1) It seems like the optimized perturbation vector \eta is still of continuous values (rather than being discretized after optimization) even for datasets of discretized node features. (2) The submission states that “In Cora, SINGLE perturbed 717 elements on average, which are 50%...” However, 50% of ALL features seems to be a bit too unrealistic if most of the normal nodes have far fewer than 50% features that are not zeros.

W3: The submission is mostly about new insights. It, however, barely invent any novel techniques. The submission would be stronger if it had explored how to improve the effectiveness of the single-node attack on the discretized datasets or against GNNs with attention.

---

> ### Author Response · Authors · 2020-11-18
> **Thank you for your review**
>
> Thank you for taking the time to review our paper! We were happy to read that you found our insights interestings and our paper friendly.
> Please see our detailed response below.
>
> >The empirical results are mostly sufficient to support the main insight that a single-node attack can be effective in many cases. Still, it would be even better if Table 2 can include results for models other than the vanilla GNNs (e.g., GAT)...
> >The submission would be stronger if it had explored how to improve the effectiveness ... against GNNs with attention
> >GAT is much harder to attack…
>
> We experimented with additional GNNs: GAT, GIN, GraphSAGE, and RobustGCN.
>
> Here are some the results, we will include the entire tables in our next revision:
> When the attacker node is chosen *randomly*
>
> |                | GCN  | GAT | GIN |GraphSage |
> |----------------|------|-----|-----|----------|
> | SINGLE         | **45.5** | **35.7**| **12.3**| **57.1**|
> | SINGLE-hops    | 48.7 | **35.5**| 18.5| 60.9            |
> | EdgeGrad       | 59.7 | 64.9| 18.5| 64.2            |
>
> When the adversary can *choose* the attacker node:
>
> |                    | GCN  | GAT | GIN |GraphSage |
> |--------------------|------|-----|-----|----------|
> | SINGLE+Topology    | **5.2**  | **27.8** | **6.3** | **6.6**          |
> | SINGLE+GradChoice  | 8.5 | 36.4| 10.0  |8.2     |
> | GlobalEdgeGrad     | 15.1 | 63.5| 10.3  | 64.7   |
>
> As shown, our results are consistent across different GNN types. Specifically, GAT does *not* seem to be harder to attack than GCN when the attacker node is chosen randomly. We hypothesize that the attack perturbation can **draw more attention** to the attacker node, thus amplifying the attack.
>
> GAT *is* more robust than GCN when the adversary can choose the attacker node. In both cases, a SINGLE attack is stronger than the corresponding baselines.
>
> As requested by AnonReviewer5, we also experimented with Robust GCN (on PubMed), and found that Robust GCN is as (if not more) vulnerable as GCN to our attack:
>
> |                       | Robust GCN | GCN (as in the paper) |
> |-----------------------|------------|-----------------------|
> | Clean                 | 73.9       | 78.5                  |
> | SINGLE                | 34.3       | 45.5                  |
> | SINGLE-hops           | 29.7       | 48.7                  |
> | SINGLE-Two-attackers  | 20.0       | 29.2                  |
> | SINGLE-Direct         | 15.8        | 0.3                   |
> | SINGLE+Topology       | 72.5        | 5.2                   |
> | SINGLE+GradChoice     | 19.6       | 8.5                   |
>
> The surprising results are that Robust GCN (Zügner & Günnemann) is vulnerable to the SINGLE attack as a standard GCN.
>
> > The attack is effective only if the attack node is close enough to the victim node
>
> This is a general limitation of GNNs - the classification of a node depends only on its neighborhood of radius L (the number of layers).
> Thus, attackers that their distance to the victim nodes is greater than the number of layers, are not considered by the victim node.
>
> >It seems like the optimized perturbation vector \eta is still of continuous values (rather than being discretized after optimization) even for datasets of discretized node features.
>
> In datasets of discrete node features - the $\eta$ vector **is** discretized after optimization.
> We will clarify this in our next revision.
>
> >The attack is effective only if the node features are of the continuous kind...
>
> As Table 1 shows, the attack is more effective than EdgeGrad when the attacker node is chosen randomly.
> As Table 2 shows, when the adversary can *choose* the attacker node, GlobalEdgeGrad is more effective in Cora and CiteSeer. However, we ran additional sets of experiments that show that in GAT and GraphSAGE -- SINGLE+GradChoice **is more effective** than GlobalEdgeGrad across all datasets (*including discrete datasets*), even in the scenario where the adversary can choose the attacker node.
> Will include these results in our next revision.

---

### Official Review · AnonReviewer2 · 2020-10-29
**Official Blind Review #2**

**Rating:** 6
**Confidence:** 4

**Review:**

The paper studies the problem of adversarial attacks in graph neural networks. It proposes a new attack strategy called single-node attack where only one node is perturbed.  A gradient-based attack algorithm is proposed to modify features of the attack node and achieve the single-node attack. Experiments have demonstrated the effectiveness of the proposed attack algorithm.
The studied problem (adversarial attack on graphs) is important and the single-node attack setting is interesting. However, there are several concerns that need to be addressed.
1.	For the constraint of unnoticeable perturbation discussed in Section 3.2, the authors argue that the perturbation would be unnoticeable if the frequency of some words is slightly modified. This contradicts with the claim that the attack should preserve the feature co-occurrence to make the perturbation unnoticeable in Nettack[1]. According to Section 4.1, SINGLE changes 50%, 31% of the node attributes on Cora and CiteSeer, respectively. With so many perturbed elements, the attack could be easily detected if many pairs of features that never appear together suddenly co-occur in a single node.
2.	Also, it is hard to evaluate whether comparing single-node attack with single-edge attack is fair, since single-edge attack only changes one edge while single-node attack can change a relative larger number of elements in the feature vector. (In Nettack, one element change in the feature matrix or adjacency matrix is considered as one perturbation) It would be more convincing if the number of elements in the feature vector changed by SINGLE is much less.
3.	For the first claim in Section 3.2 "Perturbing nodes instead of edges", although this claim shows perturbing features could be better than removing edges, it remains a question whether it is better than adding edges. Also, it might be interesting to include one more baseline where we set $\eta=-x_a$ as the feature perturbations to illustrate SINGLE can find better perturbations than simply removing the edges of the attacker node.
4.	For datasets with discrete features, SINGLE only uses "$\epsilon_{\infty}=1$" as the constraint but the discrete features are originally binary values and "$\epsilon_{\infty}=1$". This basically means no unnoticeable constraints are applied on the attacker for those datasets (Cora and CiteSeer). Hence, it would be more convincing to conduct experiments on how the model performs when using different norms as constraints (e.g., l1 and l2 norm) for discrete datasets.
5.	Since the authors claim a single-node attack is as effective as a multiple-node attack, multiple-node attacks, i.e setting the number of attacker nodes to a larger value, are suggested to be included as baselines.
6.	Given the superiority of only changing the features for one attacker node, injecting one attacker node would be of more interest since it is even more practical than modifying features on the existing node (e.g., creating an attacker account on Twitter). So, it would be better to add the experiment of node injecting to further improve the algorithm.
Minor Comments
1.	It would be better to report the standard deviation of the proposed methods in Table 1/2 considering the performances of other baselines are reported with standard deviation.
[1] Adversarial Attacks on Neural Networks for Graph Data. KDD'18

---

> ### Author Response · Authors · 2020-11-18
> **Thank you for your review (1/2)**
>
> Thank you for taking the time to review our paper! You raise important points that we think are addressable within the discussion phase.
> Please see our detailed response below.
>
> > Zügner et al. (KDD'18) claims that the attack should preserve the feature co-occurrence to make the perturbation unnoticeable
>
> Preserving feature co-occurrence is one aspect of keeping the perturbation unnoticeable.
>
> However, we believe that *only* preserving feature co-occurrence is not enough to make the attack realistic and control its noticeability, if the perturbation allows perturbing **multiple** nodes, perturbing edges **across the entire graph**, and perturbing the "victim" node itself (the "direct" attack of Zügner et al.).
>
> In this paper, we thus focus on other unnoticeability considerations (single attacker node, indirect attack, where the attacker cannot choose the specific attacker node).
>
> >SINGLE changes 50%, 31% of the node attributes on Cora and CiteSeer
>
> In *continuous datasets*, the percentage of perturbed words is much smaller: 15% in PubMed, and 9% in Twitter, while we limit the $\epsilon_{\infty}$ parameter to the small value of 0.1.
>
> If we allow using $\epsilon_{\infty} = 0.5$, this only requires perturbing 3% of the attributes on average to achieve the same effectiveness.
> If we allow using $\epsilon_{\infty} = 1$, this only requires perturbing 1.6% of the attributes on average to achieve the same effectiveness (in PubMed).
>
>
>
> > A single-node attack can change many elements in the feature vector, while a single-edge attack can change only a single edge
>
> We strengthened the EdgeGrad attack in a new approach called MultiEdgeGrad. MultiEdgeGrad can add and remove *multiple* edges that are connected to the attacker node $a$. Accordingly, a new approach called MultiGlobalEdgeGrad is equivalent to GlobalEdgeGrad, except that MultiGlobalEdgeGrad can *choose* the attacker node, and then multiple edges to add and remove. Here are the results on PubMed, using GCN:
>
> |                   | When the attacker node is chosen *randomly*      |
> |-------------------|----------|
> | EdgeGrad          | 65.1     |
> | MultiEdgeGrad     | 64.5     |
> | SINGLE            | 45.5     |
> | SINGLE-hops       | 48.7     |
>
> As shown, allowing the attacker node to add and remove multiple edges (MultiEdgeGrad) results in a very minor improvement compared to EdgeGrad, while SINGLE is much more effective.
>
> |               | When the adversary can *choose* the attacker node |
> |-------------------|----------|
> | SINGLE+Topology   | 5.2      |
> | SINGLE+GradChoice | 8.5      |
> | GlobalEdgeGrad    | 15.3     |
> | MultiGlobalEdgeGrad | 15.3     |
>
> As shown, allowing the attacker node to choose the attacker node, and then add and remove multiple edges (MultiGlobalEdgeGrad) results in a very minor improvement compared to GlobalEdgeGrad, while SINGLE+Topology and SINGLE+GradChoice are much more effective.
>
> We will include these results in our next revision.
>
>
> > Perturbing features is better than removing edges, but is it better than *adding* edges?
>
> Yes, the EdgeGrad, GlobalEdgeGrad, MultiEdgeGrad, and MultiGlobalEdgeGrad baselines can also **add edges**. We implemented this by adding all possible effective edges with weights of zero and all existing edges with weights of one, and differentiated with respect to the edge weights.
>
> We will clarify this in the paper.
>
> (see our follow-up comment)

---

> > ### Author Response · Authors · 2020-11-18
> > **Thank you for your review (2/2)**
> >
> > > It will be interesting to include a baseline where we set $\eta=-x_a$ as the feature perturbation,  to illustrate SINGLE can find better perturbations than simply removing the edges of the attacker node
> >
> > We implemented and experimented with this suggested attack, where the noise ($\eta$) simply *cancels* the node feature vector, making the new vector a vector of zeros (and thus effectively removes the edges of the attacker node in GCN):
> >
> >
> > |                          | PubMed |
> > |--------------------------|--------|
> > | Clean                    |  78.5   |
> > | SINGLE (as in Table 1)   | 45.5   |
> > | Zero features ($\eta=-x_a$) |    76.6        |
> >
> > As shown, this suggested attack is barely effective (compared to "Clean"), and indeed SINGLE can find much better perturbations.
> >
> > > it would be more convincing to conduct experiments on how the model performs when using different norms as constraints (e.g., l1 and l2 norm) for discrete datasets.
> >
> > In discrete datasets, the L1 and L2 norms are equivalent to (or correlated with) the number of perturbed vector elements, which we already measure. In discrete datasets - the only possible perturbation is "flipping" vector elements from zero to one or vice-versa.
> > > What would happen if we attack with a larger number of attackers?
> >
> > Multi-node attacks are now included and results are attached below.
> >
> > |            | 1    | 2    | 3    | 4    | 5    |
> > |------------|------|------|------|------|------|
> > | Number of attackers | 45.5 | 29.2 | 19.0 | 15.3 | 12.9 |
> >
> > As expected, allowing a larger number of attackers reduces the test accuracy. However, the main observation in this paper is that even a *single* attacker node is surprisingly effective.
> >
> > We will include these results in our next revision.
> >
> > > **Injecting** a single attacker node would be of more interest and it is more practical
> >
> > We implemented and experimented with *injecting* a single attacker that is a neighbor of the victim node. This allows to reduce the test accuracy **down to 2%** on PubMed.
> >
> > We will include this result in our next revision.
> >
> > > It would be better to report the standard deviation of the proposed methods in Table 1/2
> >
> > We ran all experiments 5 times with different random seeds, and will include standard deviation in our next revision.

---

> > > ### Comment · AnonReviewer2 · 2020-11-22
> > > **Thank you for your feedback and additional experiments.**
> > >
> > > Thank you for your feedback and additional experiments. There is still one major concern that hasn't been addressed: for discrete datasets with binary feature values, the constraint $\epsilon_{\infty}=1$  means nothing for the attacker. As a result, the algorithm tends to perturb a large portion of attributes (50% on Cora). That's why I suggest further constrain the algorithm with $L_1$ or $L_2$ norm (number of perturbed attributes).

---

> > > > ### Author Response · Authors · 2020-11-23
> > > > **Limiting the maximal L1 norm**
> > > >
> > > > We agree - in discrete datasets with binary feature values, the constraint $\epsilon_{\infty}=1$ simply means that the attacker can "flip" vector attributes.
> > > >
> > > > Following your suggestion, we performed additional experiments on Cora and CiteSeer that analyze the test accuracy on discrete datasets when we limit the allowed L1 norm of the perturbation (which is equal to L0, the number of perturbed attributes).
> > > >
> > > > The columns in the following table are the fraction of allowed perturbed attributes, and the numbers are the test accuracies for each discrete dataset:
> > > >
> > > > |              | 0% (no attack) | 10% | 20% | 30% | 40% | 50% | 60% | 70% | 80% | 90% | 100%|
> > > > |--------------|----------------|-----|-----|-----|-----|-----|-----|-----|-----|-----|-----|
> > > > | **Cora:**    |                |     |     |     |     |     |     |     |     |     |     |
> > > > | SINGLE       |  0.81          | 0.79| 0.78| 0.77| 0.75| 0.72| 0.70| 0.68| 0.65| 0.63| 0.60|
> > > > | SINGLE+GradChoice|  0.81      | 0.76| 0.74| 0.71| 0.65| 0.59| 0.53| 0.48| 0.44| 0.39| 0.31|
> > > > | **CiteSeer:** |                |     |     |     |     |     |     |     |     |     |     |
> > > > | SINGLE       |  0.69          | 0.66| 0.63| 0.56| 0.48| 0.45| 0.42| 0.40| 0.37| 0.35| 0.34|
> > > > | SINGLE+GradChoice| 0.69       | 0.64| 0.61| 0.51| 0.42| 0.35| 0.31| 0.28| 0.23| 0.20| 0.19|
> > > >
> > > >
> > > > As shown, when the allowed L1 norm is 0, there are no perturbations allowed, and the test accuracy is the "clean accuracy" (there is no attack). As we allow larger values of the L1 norm, test accuracy decreases gradually.
> > > >
> > > > It is important to note that in practice, the average number of perturbed features is *much lower* than the maximal number of allowed features. For example, in Cora, allowing 100% of the features results in *actually using only* 50% on average.
> > > >
> > > > We will include these experiments as plots in Appendix A.6 and Figure A.1 shortly.

---

### Official Review · AnonReviewer5 · 2020-11-05
**the paper lacks novelty**

**Rating:** 5
**Confidence:** 5

**Review:**

In this paper, the authors mainly show that the adversary can force the GNN to classify any target node to a chosen label by perturbing another single arbitrary node’s feature in the graph. The paper is well written and easy to understand. However, there are several concerns about the paper:

1. The novelty of the paper is rather limited. The paper simply uses the gradient attacker method to add continuous perturbations to the node attributes. What is the novelty here compared to other gradient based attacks for the data without graph structure such as images? I don’t see any novelty or contribution from the methodology perspective.

2. The problem setting needs to be well discussed. In the introduction, the authors use the example of crafting adversarial posts to motivate the problem. However, in the problem definition, it becomes adding perturbations to the node features.  Modifying the node’s feature in realistic settings is even harder than modifying the graph structure to me .(They could revise the words or sentences in the post, but they could not add the feature vector directly as the feature vector are usually preprocessed by some other models).

3. The authors should include some robust GNNs such as [1][2] as baselines to test how effective is the proposed method. The authors should also consider add more baselines as the attack methods.

[1] Zügner, Daniel, and Stephan Günnemann. "Certifiable robustness and robust training for graph convolutional networks." In Proceedings of the 25th ACM SIGKDD International Conference on Knowledge Discovery & Data Mining, pp. 246-256. 2019.
[2] Jin, Hongwei, and Xinhua Zhang. "Robust Training of Graph Convolutional Networks via Latent Perturbation." ECML-PKDD 2020

---

> ### Author Response · Authors · 2020-11-18
> **Thank you for your review**
>
> Thank you for taking the time to review our paper! You raise important points that we think are addressable within the discussion phase.
> Please see our detailed response below.
>
> > What is the novelty here?
>
> The novelty in this paper is the *observation* and *discovery* that a single-node attack is effective and harmful, in contrast with some of the conclusions of previous work, while "the attacking scenario in this paper is more realistic" (AnonReviewer4). As AnonReviewer3 wrote, we believe that this discovery "*can potentially complement our understanding of GNNs and bring novel ideas in the future*".
> > How can you perturb node features by crafting adversarial Twitter posts?
>
> In the Twitter dataset (Ribeiro et al., 2017,2018), a user's node feature vector is the average of the frequencies of the user's tweets, where every tweet is a vector of its word frequencies. Thus, crafting a new adversarial tweet affects the user's node features.
> We will clarify this in our next revision.
>
>
> > The authors should include some robust GNNs such as [1][2] to test how effective is the proposed method
>
> Robust GCN is now included in our baselines with the following results (on PubMed):
>
> |                       | Robust GCN | GCN (as in the paper) |
> |-----------------------|------------|-----------------------|
> | Clean                 | 73.9       | 78.5                  |
> | SINGLE                | 34.3       | 45.5                  |
> | SINGLE-hops           | 29.7       | 48.7                  |
> | SINGLE-Two-attackers  | 20.0       | 29.2                  |
> | SINGLE-Direct         | 15.8        | 0.3                   |
> | SINGLE+Topology       | 72.5        | 5.2                   |
> | SINGLE+GradChoice     | 19.6       | 8.5                   |
>
> The surprising results are that Robust GCN ([1]) is  vulnerable to the SINGLE attack as a standard GCN. This shows that there is still much room for novel ideas and improvements to the robustness of current GNNs.
> We will include these results in our next revision.

---

### Author Response · Authors · 2020-11-20
**To all reviewers: a summary of changes in the revised version**

We would like to thank all of the reviewers for taking the time and effort to review our paper!
We really feel that their useful comments have helped us improve the paper.

We updated our submission and included the following main changes:

1. All experiments now include standard deviations (as commented by AnonReviewer2).

2. We included a new section that investigates to what extent can *adversarial training* defend against SINGLE (Section 4.4, Table 4). The main results are that adversarial training does improve the robustness of the model, but the attack is still very effective:

|                       | Standard Training | Adversarial Training |
|-----------------------|------------|-----------------------|
| Clean (no attack)     | 78.5       | 76.9                  |
| SINGLE                | 45.5       | 58.5                  |
| SINGLE-hops           | 48.7       | 62.1                  |
| SINGLE-Two-attackers  | 27.7       | 40.7                  |
| SINGLE-Direct         | 0.3        | 4.6                   |
| SINGLE+GradChoice     | 8.5        | 30.6                  |
| SINGLE+Topology       | 5.2        | 21.1                  |


3. We included more experiments showing the effectiveness of SINGLE on different GNN types:  GAT, GIN, GraphSage, SGC and Robust GCN, across multiple datasets and multiple approaches, and across targeted and untargeted attacks (following the questions of AnonReviewers 3,4,5). These results are presented in Appendix A.2. Surprisingly, we find that Robust GCN (Zügner & Günnemann 2019) is as vulnerable to the SINGLE attack as a standard GCN, showing that there is still much room for novel ideas and improvements to the robustness of current GNNs.

4. Following the question of AnonReviewers 2,4, we explored the relation between the value of $\epsilon_{\infty}$ and the number of perturbed nodes: if we allow using $\epsilon_{\infty} = 0.5$, this only requires perturbing 3% of the attributes on average to achieve the same effectiveness. If we allow using $\epsilon_{\infty} = 1$, this only requires perturbing 1.6% of the attributes on average to achieve the same effectiveness (in PubMed). These results are presented in Section 4.1.


5. Following the comments of AnonReviewers 1,2 we included two new baselines that allow adding and removing *multiple nodes that belong to the same attacker* (Appendix A.4 and Table A.10). *MultiEdgeGrad* can add and remove multiple edges that are connected to the attacker node $a$. Accordingly, a new approach called MultiGlobalEdgeGrad which is the equivalent of multiple edges attack using GlobalEdgeGrad. We show that allowing the attacker to add and remove multiple edges (MultiEdgeGrad) results in a very minor improvement compared to EdgeGrad, while SINGLE is much more effective.

6. Following the suggestion of AnonReviewer2, we included a baseline attack called *Zero features* (Appendix A.5.1 and Table A.11) that simply zeros the node feature vector, making the new vector a vector of zeros. We found this baseline attack to be barely effective, and that SINGLE can find much better perturbations:

|                          | PubMed |
|--------------------------|--------|
| Clean                    |  78.5   |
| SINGLE (as in Table 1)   | 45.5   |
| Zero features ($\eta=-x_a$) |    76.6        |

7. Following the suggestions of AnonReviewers 1,2, we included a baseline *injection* attack, where we inject a single attacker. This reduces the test accuracy **down to almost 0%** (Appendix A.5.2 and Table A.12).

8. Following the question of AnonReviewers 1,2, we explored the effect of the number of attackers on the test accuracy (Appendix A.5.3). As expected, allowing a larger number of attackers reduces the test accuracy. However, the main observation in this paper is that even a single attacker node is surprisingly effective.

9. Following the suggestion of AnonReviewer4, we clarified which one is the "attacker node" and which one is the "victim node" in Figure 1.

10. **Our code is now publicly available** (anonymously) at https://github.com/gnnattack/SINGLE

Please let us know if there are any additional questions, would be very happy to do any follow-up discussion or address any additional comments.

---

### Author Response · Authors · 2020-11-24
**To all reviewers: submission update, 11/24/2020**

We thank again the reviewers for their useful comments and suggestions!

Following the suggestion of AnonReviewer2, we performed additional experiments on Cora and CiteSeer that analyze the test accuracy on discrete datasets when we limit the allowed L1 norm of the perturbation (which is equal to L0, the number of perturbed attributes).

|              | 0% (no attack) | 10% | 20% | 30% | 40% | 50% | 60% | 70% | 80% | 90% | 100%|
|--------------|----------------|-----|-----|-----|-----|-----|-----|-----|-----|-----|-----|
| **Cora:**     |                |     |     |     |     |     |     |     |     |     |     |
| SINGLE       |  0.81          | 0.79| 0.78| 0.77| 0.75| 0.72| 0.70| 0.68| 0.65| 0.63| 0.60|
| SINGLE+GradChoice|  0.81      | 0.76| 0.74| 0.71| 0.65| 0.59| 0.53| 0.48| 0.44| 0.39| 0.31|
| **CiteSeer:** |                |     |     |     |     |     |     |     |     |     |     |
| SINGLE       |  0.69          | 0.66| 0.63| 0.56| 0.48| 0.45| 0.42| 0.40| 0.37| 0.35| 0.34|
| SINGLE+GradChoice| 0.69       | 0.64| 0.61| 0.51| 0.42| 0.35| 0.31| 0.28| 0.23| 0.20| 0.19|


As we allow larger values of the L1 norm, test accuracy decreases gradually.
In practice, the average number of perturbed features is *much lower* than the maximal number of allowed features. For example, in Cora, allowing 100% of the features results in *actually using only* 50% on average.

We updated our submission and included these experiments in Appendix A.6 and Figure A.1.

---

### Decision · Program_Chairs · 2021-01-07
**Final Decision**

**Decision:**

Reject

**Comment:**

The paper deals with adversarial attacks on graph neural networks, a new and promising field in graph representation learning. The paper analyzes a new extreme setting of attack for a single node, and presents important insights, albeit not new algorithms.

The reviewers were not particularly enthusiastic and complained about
- limited novelty in light of Zuegner et al
- missing baselines
- doubts about the attack setting with a selected attacker node

The authors provided an elaborate rebuttal, including specific responses to the above questions, however, the final scores are not quite above the bar, especially having in mind the sheer number of submissions on graph deep learning this year. We, therefore, recommend rejection and encourage the author to publish the paper elsewhere.